# Essential medicines and technology for hypertension in primary healthcare facilities in Ebonyi State, Nigeria

**Azuka Stephen Adeke**[1,2]*, **Chukwuma David Umeokonkwo**[1,2], **Muhammad Shakir Balogun**[1,3], **Augustine Nonso Odili**[4]

1 Nigeria Field Epidemiology and Laboratory Training Program, Abuja, Nigeria, 2 Department of Community Medicine, Alex Ekwueme Federal University Teaching Hospital, Abakaliki, Ebonyi State, Nigeria, 3 African Field Epidemiology Network, Abuja, Nigeria, 4 Circulatory Health Research Laboratory, College of Health Sciences, University of Abuja, Abuja, Nigeria

* azukaadeke@gmail.com

## Abstract

### Introduction

Cardiovascular diseases (CVDs) now constitute major cause of morbidity and mortality in middle and low income countries including Nigeria. One of the major efforts at controlling CVDs in Nigeria includes expanding universal access to care through the primary healthcare (PHC) system. The study was to assess essential medicines and technology for control of hypertension in PHC facilities in Ebonyi Nigeria.

### Methods

The study used mixed method cross-sectional survey to assess availability, affordability and accessibility of essential medicines and technology in 45 facilities and among their patients with hypertension (145).

### Results

Most of the PHC facilities (71.1%) assessed were rural. The heads of facilities were mainly community health extension workers (86.7%). One (2.2%) facility had a pharmacy technician. All facilities had been supervised by the regulatory authority in the last one year. Out of 15 anti-hypertensive drugs assessed, 10 were available in some facilities (2.2%–44.4%) but essential drug availability was low (<80%). Only mercury sphygmomanometers were available in all facilities with 82.2% functioning. Stethoscopes were available in all facilities and 95.6% were functional. Glucometers were present in 20.0% of facilities and were all functional.

All the respondents (100.0%) reported they could not afford their anti-hypertensive drugs. Median monthly income was 8,000 Nigerian Naira (range = 2,000–52,000). Median monthly cost of anti-hypertensive drugs was 3,500 Naira (range = 1,500–10,000). For 99 (68.3%) of the respondents, the facilities were accessible. Median cost of transportation for care was 400 Naira (range = 100–2,000).

**Data Availability Statement:** All relevant data are within the manuscript and its Supporting information files.

**Funding:** ASA received funding from the Training Programs in Epidemiology and Public Health Interventions Network (TEPHINET, https://www.tephinet.org/). The funders had no role in study design, data collection and analysis, decision to publish, or preparation of the manuscript.

**Competing interests:** The authors have declared that no competing interests exist.

## Conclusion

Gaps still exist in the provision of hypertension control services in PHC facilities in Ebonyi State, Nigeria. The essential drugs were not always available, and cost of the drugs were still a challenge to the patients. There is urgent need to expand health insurance coverage to rural communities to ameliorate the catastrophic out-of-pocket health expenditures and improve control of CVDs.

## Introduction

Worldwide, hypertension is estimated to cause up to 7.5 million deaths, about 12.8% of the total of all deaths [1]. According to the Global Health Observatory data, across the World Health Organization (WHO) regions, the prevalence of raised blood pressure was highest in Africa, where it was 46% for both sexes combined. Both men and women have high rates of raised blood pressure in the Africa region, with prevalence rates over 40% [1]. In Nigeria, a random-effects meta-analysis estimated overall prevalence of hypertension as 28.9% with a prevalence of 29.5% among men and 25.0% among women [2]. Prevalence of hypertension may be higher in urban settings as shown by a Nigerian study with overall prevalence ranging from 17.5% to 51.6% in urban areas and 4.6% to 43% in rural areas [3]. However, using a current American College of Cardiology/American Heart Association 2017 guidelines, a study in an urban setting of Nigeria showed a prevalence of 56.5% [4]. Across the country, the south-eastern zone (which includes Ebonyi State) was recently found to have a higher prevalence (52.8%) of hypertension compared to other parts of the country [5].

Despite the high burden of hypertension in Nigeria, many clients who use primary healthcare facilities may have hypertension or significant cardiovascular disease risk that goes undetected and untreated [6]. This is on a background where in many regions of Nigeria, the government-owned primary healthcare facilities are usually the only source of formal healthcare available. It has been identified in a previous study that essential medicines were not readily available in some primary healthcare centres and equipment were either lacking or not optimally functional [7]. This presupposes that majority of the healthcare facilities do not meet the minimum standard for primary healthcare service delivery, although, Ebonyi State was not one of the States surveyed.

Due to the drive to push universal health coverage, Ebonyi State Government in Nigeria set up Ebonyi State Health Insurance Agency to improve access and utilization of health services especially through primary and secondary healthcare facilities. The State also established Ebonyi State Primary Heath Care Development Agency to strengthen the organization and functionality of primary healthcare structure in the State. Health insurance packages being planned, include management of hypertension. The objective of this study was to evaluate the essential medicines and technology for control of hypertension in primary healthcare facilities in selected Local Government Areas (LGAs) of Ebonyi State, Nigeria. The findings from this study might be vital in strengthening health systems at the primary healthcare level to improve universal health coverage.

## Materials and methods

### Study area

The study was conducted in Ebonyi State, one of the south-eastern States of Nigeria, between January and July 2020. Ebonyi State is made of 13 LGAs, ten of which are largely rural. The main stay of the state economy is agriculture and largely unorganized solid mineral mining.

The people are largely farmers especially rice and yam farming. The projected population of the state as at 2019 is put at 3,112,220. It has relatively young population with over 60% of the population being less than 50 years. The health care delivery system is largely provided by government in a hierarchical manner: primary, secondary and tertiary hospitals. Private and faith-based organizations also provide health services at primary and secondary level. The state is served with two tertiary hospitals, thirteen secondary level general hospitals and 417 primary healthcare facilities provided and supported by government. The health service provision at the primary healthcare facilities are mainly headed by nurses, community health extension workers (CHEWs) and community health officers due to limited number of doctors.

As observed in most states in the country, health service delivery in Ebonyi State is structured into a three-tier system consisting of the primary, secondary and tertiary healthcare levels. While the federal government has the responsibility for the tertiary healthcare, the state government is responsible for the secondary healthcare and the local government in charge of primary healthcare with support of state government. However, in 2010, it was noted that the health system in the state was extremely weak with the primary and secondary health care levels virtually collapsed [8]. Hence, there was need for the establishment of Ebonyi State Primary Heath Care Development Agency and Ebonyi State Health Insurance Agency to strengthen healthcare at the grassroot through primary healthcare system as the easiest means of delivering basic healthcare services across the state.

## Study population

The study population comprised primary healthcare facilities in selected local government areas and people with hypertension who access these facilities.

## Study design

The study used mixed method comprising descriptive cross-sectional design and focus group discussions.

## Sampling procedure/sample size

Facilities were selected using multistage sampling. In the first stage, the LGAs were stratified into the 3 senatorial districts that comprise the state and one LGA was selected from each senatorial district through balloting. In the second stage, in the selected LGAs, a list of all the primary healthcare facilities in each of the LGAs was obtained from the Ebonyi State Health Insurance Agency, 15 primary healthcare facilities were selected from each of the 3 LGAs through balloting method of simple random sampling. A total of 45 primary healthcare facilities were selected in all. Also, all persons with hypertension who accessed the selected facilities during the study period (January and July 2020) and gave consent to participate in the study were interviewed for this study.

## Data management

To evaluate availability of essential medicines and technology, a checklist [9] (adapted from Nigeria's Essential Medicine List) was used to assess anti-hypertensive drugs present in a health facility as well as technology (sphygmomanometer, stethoscope, test kit for urinalysis, lipid profile testing, glucometer, weighing scale, measuring tapes). Availability of test kit for urinalysis, lipid profile testing, glucometer, weighing scale, and measuring tapes was assessed due to their importance as screening tools for other common non-communicable diseases (NCDs) and their risk factors.

To assess affordability and accessibility of essential medicines and technology for hypertension, data were collected from respondents using interviewer-administered questionnaires and focus group discussions to triangulate the quantitative data. Patient registers were reviewed in the primary healthcare facilities and all those with hypertension were enrolled to assess affordability and accessibility using questionnaires. One focus group discussion was conducted in each LGA by selecting 8 to 11 participants among those with hypertension for each discussion.

Data was cleaned by checking for consistency and completeness. Data analysis was done with Statistical Package for Social Sciences (IBM SPSS) software version 25. Binary variables were described using frequencies and percentages; and continuous variables using medians and ranges. Responses of the focus group discussions were analyzed in themes.

The methodology [10] developed by the WHO and Health Action International on measuring availability, affordability and accessibility were adapted for the analysis. Availability was the proportion of primary healthcare facilities in which an essential medicine or technology was found on the day of data collection. Median availability was gotten for all essential medicines and technology. To measure affordability, at a 5% catastrophic threshold, the medicine with price $P$ was unaffordable for people earning less than 20 times $P$ where $P$ was monthly cost of medication. The proportion of respondents for which purchasing a medicine costing $P$ was catastrophic was then calculated. Accessibility was evaluated as primary healthcare facilities within one hour's walk from the homes of the respondents.

### Ethical consideration

Ethical approval was obtained from the Ethical Review Committee of Ebonyi State Ministry of Health with approval number SMOH/EC/003/2020. Respondents were informed of their voluntariness to participate in the study, and confidentiality and anonymity of data collected were maintained by avoiding inclusion of possible identifiers, such as names and contact details. Written consent was obtained from the respondents.

### Results

Forty-five primary healthcare facilities were assessed and majority of them were in rural areas (32, 71.1%), while 11 (24.4%) and 2 (4.4%) were in semi-urban and urban settings respectively. The officers-in-charge of the facilities were mainly CHEWs (39, 86.7%) with few nurses (6, 13.3%). Only 1 (2.2%) primary healthcare facility had a pharmacy technician. All the facilities had been inspected by Ebonyi State Primary Health Care Development Agency in the last one year. Out of 15 anti-hypertensive drugs assessed, 11 were available in some facilities in the following proportions: Amlodipine (20, 44.4%), Methyldopa (18,40.0%), Nifedipine (17, 37.8%), Lisinopril (13, 28.9%), Amiloride+Hydrochlorothiazide (9,20.0%), Propranolol (2, 4.4%), Losartan (2, 4.4%), Valsartan (2, 4.4%), Hydralazine (2, 4.4%), and Labetalol (1, 2.2%) (Table 1).

From the focus group discussions conducted, there were complaints of lack of antihypertensive medications in some of the facilities and could discourage the clients from visiting the health facilities.

*The drugs are not available here so checking of blood pressure is futile as we cannot get the drugs after checking the blood pressure. I have had emergencies and when rushed here, there was no medication to be given*

*(Participant, Ohaukwu Local Government Area)*

**Table 1. Availability of essential medicines for control of hypertension in primary healthcare facilities in Ebonyi State, Nigeria.**

| Variable | Overall availability in all primary healthcare facilities n = 45 (%) | Ikwo LGA facilities n = 15 (%) | Ohaukwu LGA facilities (n = 15) (%) | Onicha LGA facilities n = 15 (%) |
|---|---|---|---|---|
| Amiloride +hydrochlorothiazide | 9 (20.0) | 1 (6.7) | 2 (13.3) | 6 (40.0) |
| Amlodipine | 20 (44.4) | 2 (13.3) | 5 (33.3) | 13 (86.7) |
| Atenolol | 0 (0.0) | 0 (0.0) | 0 (0.0) | 0 (0.0) |
| Bendrofluazide | 0 (0.0) | 0 (0.0) | 0 (0.0) | 0 (0.0) |
| Captopril | 0 (0.0) | 0 (0.0) | 0 (0.0) | 0 (0.0) |
| Hydralazine | 2 (4.4) | 1 (6.7) | 0 (0.0) | 1 (6.7) |
| Labetalol | 1 (2.2) | 0 (0.0) | 0 (0.0) | 1 (6.7) |
| Lisinopril | 13 (28.9) | 2 (13.3) | 4 (26.7) | 7 (46.7) |
| Losartan | 2 (4.4) | 0 (0.0) | 0 (0.0) | 2 (13.3) |
| Methyldopa | 18 (40.0) | 1 (6.7) | 8 (53.3) | 9 (60.0) |
| Nifedipine | 17 (37.8) | 2 (13.3) | 6 (40.0) | 9 (60.0) |
| Nimodipine | 0 (0.0) | 0 (0.0) | 0 (0.0) | 0 (0.0) |
| Propranolol | 2 (4.4) | 0 (0.0) | 0 (0.0) | 2 (13.3) |
| Reserpine | 0 (0.0) | 0 (0.0) | 0 (0.0) | 0 (0.0) |
| Valsartan | 2 (4.4) | 1 (6.7) | 0 (0.0) | 1 (6.7) |

LGA—Local Government Area.

*Sometimes, I go to the health facility close to me to get my drugs but these drugs might not be available at the health facility and I am forced to go to farther pharmacies to get them which may cost about 2,400 Naira for transportation*

*(Participant, Ikwo Local Government Area)*

Among the technology assessed, mercury sphygmomanometers were available in all the facilities out of which 37 (82.2%) were functioning. There was no digital sphygmomanometer in any of the facilities. Stethoscope were available in all the facilities and were functional in 43 (95.6%) of them. Urine test kits were available and functional in 14 (31.1%) facilities. There was no lipid profile testing in any of the facilities. Glucometers were present in 9 (20.0%) facilities and were all functional. Weighing scales were available in all the facilities with 43 (95.6%) that were functional. Measuring tapes were present in 43 (95.6%) facilities and were functional (Table 2).

**Table 2. Availability of technology for control of hypertension in primary healthcare facilities in Ebonyi State, Nigeria.**

| Variable | Overall availability in all primary healthcare facilities n = 45 (%) | Ikwo LGA facilities n = 15 (%) | Ohaukwu LGA facilities (n = 15) (%) | Onicha LGA facilities n = 15 (%) |
|---|---|---|---|---|
| Sphygmomanometer | 45 (100.0) | 15 (100.0) | 15 (100.0) | 15 (100.0) |
| Stethoscope | 45 (100.0) | 15 (100.0) | 15 (100.0) | 15 (100.0) |
| Urinalysis test kit | 14 (31.1) | 3 (20.0) | 2 (13.3) | 9 (60.0) |
| Lipid profile test | 0 (0.0) | 0 (0.0) | 0 (0.0) | 0 (0.0) |
| Glucometer | 9 (20.0) | 2 (13.3) | 0 (0.0) | 7 (46.7) |
| Weighing scale | 45 (100.0) | 15 (100.0) | 15 (100.0) | 15 (100.0) |
| Measuring tape | 43 (95.6) | 15 (100.0) | 15 (100.0) | 13 (86.7) |

LGA—Local Government Area.

One hundred and forty-five respondents were identified as being hypertensive from the records of 45 primary healthcare facilities assessed. All the respondents (145, 100.0%) could not afford their antihypertensive medications as they earned less than 20 times the cost of their monthly medications. The median monthly income was 8,000 Nigerian Naira (range = 2,000–52,000). The median monthly cost of antihypertensive medications was 3,500 Nigerian Naira (range = 1,500–10,000). Responses from some participants in the qualitative component of the project buttressed the problem of unaffordability of their medications.

*Most of us are old and do not have any means of livelihood. We solely depend on our children and as such, have to wait for when they have money for our drugs.*

*(Participant, Ohaukwu Local Government Area)*

*Most of us do not have the money or means of purchasing our medications, so we depend on assistance. That is why we appreciate your coming to us as we expect you will help us get assistance of the government.*

*(Participant, Onicha Local Government Area)*

In assessing affordability of their medications, the participants were asked if there was a time in the past 12 months when they did not take their medications as prescribed because of cost.

*About 4 months ago, I went to the health center to check my blood pressure, but I did not have money to procure the drugs prescribed. On coming back to the center weeks later, the blood pressure rose so high and I still had no money to purchase the drugs and made calls until the money was provided by my son.*

*(Participant, Ohaukwu Local Government Area)*

*In the past three months, I have been off medications. I usually check my blood pressure at the nearest primary health center and if my medications finish and there is no money to purchase new ones, I stay without it.*

*(Participant, Ikwo Local Government Area)*

*Since I had a surgery, I have not been able to get my antihypertensive medications as a result of lack of money. I have not had my medication for the past 9 months and my blood pressure was last checked 9 months ago.*

*(Participant, Onicha Local Government Area)*

However, for 99 (68.3%) of the respondents, the primary healthcare facilities were accessible as they were within one hour's walk from the homes. Although, only 47 (32.4%) of the respondents mentioned that it was easy for them to get to the facility where they receive treatment for hypertension. The median cost of transportation from their houses to the facility where they receive treatment and back was 400 Nigerian Naira (range = 100–2,000). The participants in the focus group discussions acknowledged the problem of accessibility which is sometimes due to distance, bad road network, and cost of transportation to access care in the primary healthcare facilities.

*Some of us live far away from the health centers and as a result of that, have difficulty walking to them. In extreme cases, some of us feel dizzy and have to get a means of transporting ourselves to the facility to get our drugs.*

*(Participant, Ohaukwu Local Government Area)*

*We use public transportation to get to the health facility. But because there may be no drugs in the facility, we might spend about 1,,000–1,400 Naira to get to Abakaliki town in order to purchase our drugs.*

*(Participant, Ikwo Local Government Area)*

*Some of us are close to health facilities while some live afar off and spend up to 1600 Naira for each trip. For those of us staying close, motorcycle trips could range from 200–400 Naira to the facility.*

*(Participant, Onicha Local Government Area)*

## Discussion

There was low availability (less than half) of most of the anti-hypertensive medications while the others were not available in any facility. While primary healthcare in Nigeria is intended to drive healthcare services to the grassroots, the rural people tend to under-use the basic health services possibly due to some factors such as inadequacy of available services and poor availability of drugs [11]. An assessment of availability of essential NCD drugs in primary healthcare facilities in some Rwandan districts noted low availability of essential medicines for cardiovascular diseases management [12]. In many low and middle-income countries, medication supply is inconsistent [13]. WHO emphasizes the importance of strengthening cardiovascular diseases management in primary healthcare through its HEARTS technical package [14].

This study identified availability of some of the basic tools for control of hypertension (sphygmomanometer and stethoscope). However, only mercury sphygmomanometers were available, which are being discouraged from use due to some factors such as risk of mercury toxicity, reliance on the observer's skills, and calibration issues [13, 15]. The digital semi-automatic BP monitors tends to be more accurate, and hence, recommended for improved control of hypertension [16]. Most of the facilities did not have glucometer for blood sugar monitoring, despite high blood sugar (diabetes) being one of the metabolic risk factors associated with cardiovascular diseases. Diabetes and hypertension are known to share common pathways which interact and influence each other [17].

Based on the criteria to assess affordability, none of the persons with hypertension that were interviewed could afford their medications, as they would often need assistance to purchase their anti-hypertensive medications. This could commonly lead to poor adherence to their medications at times when they could not purchase or get financial support to purchase the medications with consequent poor control of blood pressure. Also, the primary healthcare facilities were accessible to most of the persons with hypertension that were interviewed. But their use of the facilities was sometimes limited by non-availability of their medications which would make them spend more to go to where they could purchase the required medications. In this study, most of the primary healthcare facilities were located in the rural areas. This emphasizes the need to strengthen the services at this level of healthcare so as to achieve universal health coverage as people living in the rural areas may not have easy access to secondary and tertiary health facilities. To improve and strengthen the services of primary healthcare, the Nigerian Government rolled out Basic Health Care Provision Fund, [18] a key component of the National Health Act, which aims to extend primary healthcare to all Nigerians by substantially increasing the level of financial resources to primary healthcare services [19]. Fifty percent of the fund will be disbursed through the National Health Insurance Scheme to its

counterparts in the states for the provision of Basic Minimum Package of Health Services, a package without charge at the point of care and include screening and facilitated referral for hypertension and diabetes [20]. Another forty-five percent of the fund shall be disbursed through the National Primary Health Care Development Agency to its counterparts in the states to strengthen primary healthcare facilities with essential medicines, vaccines and consumables, provision and maintenance of facilities, equipment and transport, and development of human resources [20].

This study shows the role of CHEWs in primary healthcare in Ebonyi State as majority of the heads of the primary healthcare facilities were CHEWs. Despite the plans of the State to achieve universal health coverage through primary healthcare, the National Standing Orders for CHEWs does not clearly state their roles in the routine control of hypertension except in cases of hypertensive emergencies in which they are instructed to refer cases [21]. There might be need to review the Standing Orders and other related policies/ guidelines such as task-shifting to improve the role of CHEWs in the control of hypertension through capacity building at their level of expertise. Also, technology may be used to improve hypertension control at primary healthcare level through channels like tele-monitoring where medical doctors can guide CHEWs on management of hypertension. A pilot study based on a pharmacy-based hypertension care model was conducted in Lagos with the use of mHealth to increase access and quality of hypertension care through task-shifting from medical doctors to pharmacists [22]. The study was shown to be a feasible care model which improved patient accessibility, attention, adherence, information provision, and BP control with the assistance of cardiologists to safeguard the quality of care provided by the pharmacists. Still in relation to technology for hypertension control, text messaging may be used to remind patients of their clinic visits and drug pick-ups, in settings where there is availability of anti-hypertensive medications.

To buttress the need for review of policies and guidelines on hypertension, the Nigerian Hypertension Society recently updated its guidelines for management of hypertension which was produced in 2020 and defined hypertension with a blood pressure cut-off of 140/90 mmHg [23]. In 2019, Nigeria's Federal Ministry of Health in collaboration with the WHO and key stakeholders launched the first National Multi-Sectoral Action Plan for the Prevention and Control of Non-Communicable Diseases for the country [24].

One of the priority actions of the National Multi-Sectoral Action Plan is to integrate NCD prevention, care and treatment into basic primary healthcare with referral to all levels of care [25]. Activities to achieve this action include reviewing existing guidelines for primary healthcare to include comprehensive NCD prevention and treatment, as well as develop service integration guidelines on NCDs. Another priority action is to build the capacity of healthcare workers on integrated management of NCDs [25]. Activities set to achieve this include expanding the task-shifting/task-sharing policy for frontline primary healthcare workers to include NCDs, and develop curriculum for in-service and pre-service training on NCD management. Another key priority action related to this study is to scale up coverage of early detection and diagnosis at primary healthcare level [25]. Activities for this action include reviewing the essential medicines list and minimum standards for primary healthcare to include essential tools for NCD diagnosis (such as glucometers and blood pressure monitors), and procurement and dissemination of these tools for NCD diagnosis.

Our study finding of low availability of anti-hypertensive medications further shows the need to review the existing guidelines to improve NCD management, especially hypertension. Our findings on headship of primary healthcare facilities being mainly CHEWs emphasizes the importance of expanding the role of CHEWs through task-shifting/task-sharing on NCD management. One of our findings on low availability of technology such as glucometers may serve as guidance for provision essential tools for NCD diagnosis.

Our study was not without limitation as the data used to assess accessibility and affordability of essential medicines were self-reported from the study respondents with hypertension and may have had recall bias.

## Conclusion

Gaps still exist in the control of hypertension in the primary healthcare facilities in Ebonyi State and this may be similar in other States of Nigeria. There is need for review of policies to improve the roles of CHEWs in the management of hypertension and appropriate referral system. Also, the judicious implementation of the Basic Health Care Provision Fund through the Ebonyi State Health Insurance Agency and Ebonyi State Primary Heath Care Development Agency in Ebonyi State might be key to the control of hypertension and other non-communicable diseases by improving availability and affordability of essential medicines.

## Supporting information

**S1 Dataset. Data on availability of essential medicines and technology for the control of hypertension in the primary healthcare facilities in Ebonyi State, Nigeria.**
(SAV)

**S2 Dataset. Data on affordability and accessibility of essential medicines and technology in the primary healthcare facilities in Ebonyi State, Nigeria among persons with hypertension.**
(SAV)

## Acknowledgments

We acknowledge Fleetwood Loustalot for his invaluable mentorship in the course of this study. We would like to thank Osarhiemen Iyare and Olaedo Nnachi for their effort and assistance with data collection.

## Author Contributions

**Conceptualization:** Azuka Stephen Adeke, Chukwuma David Umeokonkwo, Muhammad Shakir Balogun.

**Data curation:** Azuka Stephen Adeke.

**Formal analysis:** Azuka Stephen Adeke.

**Funding acquisition:** Azuka Stephen Adeke.

**Investigation:** Azuka Stephen Adeke.

**Methodology:** Azuka Stephen Adeke, Chukwuma David Umeokonkwo, Muhammad Shakir Balogun, Augustine Nonso Odili.

**Project administration:** Azuka Stephen Adeke.

**Resources:** Azuka Stephen Adeke.

**Supervision:** Augustine Nonso Odili.

**Validation:** Augustine Nonso Odili.

**Writing – original draft:** Azuka Stephen Adeke.

**Writing – review & editing:** Azuka Stephen Adeke, Chukwuma David Umeokonkwo, Muhammad Shakir Balogun, Augustine Nonso Odili.

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
