## [Decision Letter · Decision Letter 0]

18 Nov 2021

PONE-D-21-19423Essential medicines and technology for hypertension in primary healthcare facilities in Ebonyi State, NigeriaPLOS ONE

Dear Dr. Adeke,

Thank you for submitting your manuscript to PLOS ONE. After careful consideration, we feel that it has merit but does not fully meet PLOS ONE’s publication criteria as it currently stands. Therefore, we invite you to submit a revised version of the manuscript that addresses the points raised during the review process.

Please take into consideration the constructive comments from both referees.

We look forward to receiving your revised manuscript.

Kind regards,

Susan Horton

Academic Editor

PLOS ONE

Journal Requirements:

2. We note that Figure 1 in your submission contain [map/satellite] images which may be copyrighted. All PLOS content is published under the Creative Commons Attribution License (CC BY 4.0), which means that the manuscript, images, and Supporting Information files will be freely available online, and any third party is permitted to access, download, copy, distribute, and use these materials in any way, even commercially, with proper attribution. For these reasons, we cannot publish previously copyrighted maps or satellite images created using proprietary data, such as Google software (Google Maps, Street View, and Earth). For more information, see our copyright guidelines: http://journals.plos.org/plosone/s/licenses-and-copyright.

 a. You may seek permission from the original copyright holder of Figure(s) [#] to publish the content specifically under the CC BY 4.0 license. 

Additional Editor Comments (if provided):

Both reviewers feel this is a good paper on an important topic, and have provided helpful comments to strengthen the analysis. Please revise, taking these suggestions into account.

Reviewers' comments:

Reviewer's Responses to Questions

**Comments to the Author**

1. Is the manuscript technically sound, and do the data support the conclusions?

Reviewer #1: Yes

Reviewer #2: Yes

2. Has the statistical analysis been performed appropriately and rigorously? 

Reviewer #1: Yes

Reviewer #2: Yes

3. Have the authors made all data underlying the findings in their manuscript fully available?

Reviewer #1: Yes

Reviewer #2: No

4. Is the manuscript presented in an intelligible fashion and written in standard English?

Reviewer #1: Yes

Reviewer #2: Yes

5. Review Comments to the Author

Reviewer #1: This is a relatively simple, but important and well written, paper detailing the state of primary care in a state of Nigeria with respect to availability of equipment and medication. The results are grave indeed and should be considered unacceptable in contemporary times. I think it is of interest to readers and illustrates the tremendous gaps still present in economically developing nations.

I was pleased to see that funding is being directed towards alleviating the problem. Can the authors comment further on the potential for technology to assist with improvement in control and supply/distribution? Is it feasible to bring medications to patients via a mobile unit? Do patients own smartphones, like they do in many lower income areas of the world? Can BP monitoring and coaching be done via a simple digital health platform such that best practice can be automated, helping community health workers to deliver care (because the physician/nurse shortage is not easily solved).

Reviewer #2: Thank you - I enjoyed reading this paper which I believe is important given the growing rates of hypertension and CVD across Africa - now overtaking infectious diseases as the biggest killer in sub-Saharan Africa. My comments are only minor with potential references as suggestions only based on work i have published with others across Africa and Nigeria on NCDs, etc.

These include:

A) Introduction

a) Illness can have catastrophic consequences for the income of families in Nigeria - Aregbeshola BS, Khan SM. Out-of-Pocket Payments, Catastrophic Health Expenditure and Poverty Among Households in Nigeria 2010. International journal of health policy and management. 2018;7(9):798-80

b) However a concern is that prescribing physicians do not always know the costs of the medicines they prescribe (in Fadare JO, Enwere OO, Adeoti AO, Desalu OO, Godman B. Knowledge and Attitude of Physicians Towards the Cost of Commonly Prescribed Medicines: A Case Study in Three Nigerian Healthcare Facilities. Value in health regional issues. 2020;22:68-74) and there can be concerns with the quality of generic medicines in Nigeria adding to the problem of affordability (in Fadare JO, Adeoti AO, Desalu OO, Enwere OO, Makusidi AM, Ogunleye O, et al. The prescribing of generic medicines in Nigeria: knowledge, perceptions and attitudes of physicians. Expert review of pharmacoeconomics & outcomes research. 2016;16:639-50)

c) There are also concerns with the management of diabetes (with hypertension a key issue) in patients in Nigeria including usage of long-acting insulin analogues (in Godman B, Basu D, Pillay Y et al. Ongoing and planned activities to improve the management of patients with Type 1 diabetes across Africa; implications for the future. Hospital practice. 2020;48(2):51-67; Godman B, Basu D, Pillay Y, Mwita JC et al. Review of Ongoing Activities and Challenges to Improve the Care of Patients With Type 2 Diabetes Across Africa and the Implications for the Future. Frontiers in pharmacology. 2020;11(108) and Haque M, Islam S, Abubakar AR, Sani IH et al. Utilization and expenditure on long-acting insulin analogs among selected middle-income countries with high patient co-payment levels: findings and implications for the future. J Appl Pharm Sci. 2021;11(07):172–182). I mention diabetes as in table 2 you discuss lipid tests, weight/ height examinations and glucose levels.

B) Methodology

a) why when measuring hypertension did you include in Table 2 measures such as blood glucose levels, height and weight measurements as well as lipid levels - need an explanation for this

b) Focus group discussion - how was the questionnaire developed - was this based on previous literature, etc.?

c) How long do patients have to wait to see a HCW in a typical PHC - I say this because waiting times can be long which leads to patients self-purchasing their medicines from pharmacies if they can

d) Is any counselling given to patients during their visits regarding lifestyle changes/ adherence to medicines prescribed, etc.?

C) Discussion

a) I would not include Brazil as they do have universal health care - so studies undertaken to assess availability to seek to ensure this. Interestingly updates on ref 15 include (if interested): Garcia MM, Barbosa MM et al. Indicator of access to medicines in relation to the multiple dimensions of access. Journal of comparative effectiveness research. 2019;8:1027-41; Barbosa MM, Moreira TA et al. Access to medicines in the Brazilian Unified Health System's primary health care: assessment of a public policy. Journal of comparative effectiveness research. 2021;10:869-79 and Barbosa MM, Nascimento RC et al. Strategies to improve the availability of medicines in primary health care in Brazil: findings and implications. Journal of comparative effectiveness research. 2021;10:243-53

b) ref 16 is old - there have been more recent studies including e.g. Babar Z, Ramzan S et al. The Availability, Pricing, and Affordability of Essential Diabetes Medicines in 17 Low-, Middle-, and High-Income Countries. Frontiers in pharmacology. 2019;10(1375); Mukundiyukuri JP, Irakiza JJ, Nyirahabimana N et al. Availability, Costs and Stock-Outs of Essential NCD Drugs in Three Rural Rwandan Districts. Ann Glob Health. 2020;86:123 and Chow CK, Ramasundarahettige C, Hu W, AlHabib KF, Avezum A, Jr., Cheng X, et al. Availability and affordability of essential medicines for diabetes across high-income, middle-income, and low-income countries: a prospective epidemiological study. The lancet Diabetes & endocrinology. 2018;6(10):798-808 to name just a few

c) Similar comments to above - why mention diabetes when the paper is about hypertension - suggest broadening the title to talk about NCDs with a special emphasis on hypertension

d) i would like to see comments on the ways forward to achieve the goals in the recent Nigerian Plan for NCDs and how these results can be used to provide guidance to the authorities, etc.

6. PLOS authors have the option to publish the peer review history of their article (what does this mean?). If published, this will include your full peer review and any attached files.

Reviewer #1: **Yes: **Raj Padwal, University of Alberta, Edmonton, Canada

Reviewer #2: **Yes: **Brian Godman

---

## [Author Response · Author response to Decision Letter 0]

16 Jan 2022

Academic editor 

 Please ensure that your manuscript meets PLOS ONE's style requirements, including those for file naming. The PLOS ONE style templates can be found at https://journals.plos.org/plosone/s/file?id= wjVg/ PLOSOne_formatting_sample_ main_body.pdf and https://journals.plos.org/plosone/s/ file?id=ba62/PLOSOne_formatting_sample_ title_authors_affiliations.pdf

Thank you for your comments. The journal’s style requirements have been checked and adjusted made on the manuscript.

 We note that Figure 1 in your submission contain [map/satellite] images which may be copyrighted. All PLOS content is published under the Creative Commons Attribution License (CC BY 4.0), which means that the manuscript, images, and Supporting Information files will be freely available online, and any third party is permitted to access, download, copy, distribute, and use these materials in any way, even commercially, with proper attribution. For these reasons, we cannot publish previously copyrighted maps or satellite images created using proprietary data, such as Google software (Google Maps, Street View, and Earth). For more information, see our copyright guidelines: http://journals.plos.org/ plosone/s/ licenses-and-copyright.

We require you to either (1) present written permission from the copyright holder to publish these figures specifically under the CC BY 4.0 license, or (2) remove the figures from your submission:….. Thank you for the information. We have removed the figure from the manuscript.

 Please review your reference list to ensure that it is complete and correct. If you have cited papers that have been retracted, please include the rationale for doing so in the manuscript text, or remove these references and replace them with relevant current references. Any changes to the reference list should be mentioned in the rebuttal letter that accompanies your revised manuscript. If you need to cite a retracted article, indicate the article’s retracted status in the References list and also include a citation and full reference for the retraction notice. We have reviewed the reference list and corrected noted errors while adding new references as suggested by Reviewer #2.

Reviewer #1 

 This is a relatively simple, but important and well written, paper detailing the state of primary care in a state of Nigeria with respect to availability of equipment and medication. The results are grave indeed and should be considered unacceptable in contemporary times. I think it is of interest to readers and illustrates the tremendous gaps still present in economically developing nations. Thank you for your comments.

 I was pleased to see that funding is being directed towards alleviating the problem. Can the authors comment further on the potential for technology to assist with improvement in control and supply/distribution? Is it feasible to bring medications to patients via a mobile unit? Do patients own smartphones, like they do in many lower income areas of the world? Can BP monitoring and coaching be done via a simple digital health platform such that best practice can be automated, helping community health workers to deliver care (because the physician/nurse shortage is not easily solved). Thank you for the vital questions/suggestions raised. We are convinced of the potential role of technology to improve control of hypertension. These comments have been addressed in the discussion section and reads as…..“Also, technology may be used to improve hypertension control at primary healthcare level through channels like tele-monitoring where medical doctors can guide CHEWs on management of hypertension. A pilot study based on a pharmacy-based hypertension care model was conducted in Lagos with the use of mHealth to increase access and quality of hypertension care through task-shifting from medical doctors to pharmacists [23]. The study was shown to be a feasible care model which improved patient accessibility, attention, adherence, information provision, and BP control with the assistance of cardiologists to safeguard the quality of care provided by the pharmacists. Still in relation to technology for hypertension control, text messaging may be used to remind patients of their clinic visits and drug pick-ups, in settings where there is availability of anti-hypertensive medications.”

Reviewer #2 

 Thank you - I enjoyed reading this paper which I believe is important given the growing rates of hypertension and CVD across Africa - now overtaking infectious diseases as the biggest killer in sub-Saharan Africa. My comments are only minor with potential references as suggestions only based on work i have published with others across Africa and Nigeria on NCDs, etc. Thank you for your comments and suggestions. The suggestions have been incorporated to further enrich the work. 

Introduction Illness can have catastrophic consequences for the income of families in Nigeria - Aregbeshola BS, Khan SM. Out-of-Pocket Payments, Catastrophic Health Expenditure and Poverty Among Households in Nigeria 2010. International journal of health policy and management. 2018;7(9):798-80 Thank you again for this suggested citation. The authors have read through the work however, it is our considered opinion that work is not within the scope of our study. 

 However a concern is that prescribing physicians do not always know the costs of the medicines they prescribe (in Fadare JO, Enwere OO, Adeoti AO, Desalu OO, Godman B. Knowledge and Attitude of Physicians Towards the Cost of Commonly Prescribed Medicines: A Case Study in Three Nigerian Healthcare Facilities. Value in health regional issues. 2020;22:68-74) and there can be concerns with the quality of generic medicines in Nigeria adding to the problem of affordability (in Fadare JO, Adeoti AO, Desalu OO, Enwere OO, Makusidi AM, Ogunleye O, et al. The prescribing of generic medicines in Nigeria: knowledge, perceptions and attitudes of physicians. Expert review of pharmacoeconomics & outcomes research. 2016;16:639-50) The authors note that these are very important references in the Nigerian context. However, we do not want to expand the introduction with information on cost and quality of medicines.

 There are also concerns with the management of diabetes (with hypertension a key issue) in patients in Nigeria including usage of long-acting insulin analogues (in Godman B, Basu D, Pillay Y et al. Ongoing and planned activities to improve the management of patients with Type 1 diabetes across Africa; implications for the future. Hospital practice. 2020;48(2):51-67; Godman B, Basu D, Pillay Y, Mwita JC et al. Review of Ongoing Activities and Challenges to Improve the Care of Patients With Type 2 Diabetes Across Africa and the Implications for the Future. Frontiers in pharmacology. 2020;11(108) and Haque M, Islam S, Abubakar AR, Sani IH et al. Utilization and expenditure on long-acting insulin analogs among selected middle-income countries with high patient co-payment levels: findings and implications for the future. J Appl Pharm Sci. 2021;11(07):172–182). I mention diabetes as in table 2 you discuss lipid tests, weight/ height examinations and glucose levels. The authors appreciate the references suggested on diabetes. However, we kindly prefer to focus on hypertension for the introduction.

Methodology why when measuring hypertension did you include in Table 2 measures such as blood glucose levels, height and weight measurements as well as lipid levels - need an explanation for this We assessed availability of other testing tools due to their importance as screening tools for other NCDs. The following statement has been added to the “data management” subsection of the methodology as explanation: “Availability of test kit for urinalysis, lipid profile testing, glucometer, weighing scale, and measuring tapes was assessed due to their importance as screening tools for other common NCDs and their risk factors.”

 Focus group discussion - how was the questionnaire developed - was this based on previous literature, etc.? The guide used for focus group discussion was developed by authors from their work experience.

 How long do patients have to wait to see a HCW in a typical PHC - I say this because waiting times can be long which leads to patients self-purchasing their medicines from pharmacies if they can PHCs in the study area do not usually have many patients so the clients do not experience long waiting time to see a HCW except on special days for vaccination and antenatal care. Long waiting time in the study area is commonly experience in secondary and tertiary hospitals.

 Is any counselling given to patients during their visits regarding lifestyle changes/ adherence to medicines prescribed, etc.? Counselling is given to patients. However, there may be need for more training on counselling for CHEWs to improve their delivery.

Discussion I would not include Brazil as they do have universal health care - so studies undertaken to assess availability to seek to ensure this. Interestingly updates on ref 15 include (if interested): Garcia MM, Barbosa MM et al. Indicator of access to medicines in relation to the multiple dimensions of access. Journal of comparative effectiveness research. 2019;8:1027-41; Barbosa MM, Moreira TA et al. Access to medicines in the Brazilian Unified Health System's primary health care: assessment of a public policy. Journal of comparative effectiveness research. 2021;10:869-79 and Barbosa MM, Nascimento RC et al. Strategies to improve the availability of medicines in primary health care in Brazil: findings and implications. Journal of comparative effectiveness research. 2021;10:243-53 Thank you for the wonderful references suggested. Ref 15 from Brazil has been removed since they now offer universal healthcare. We also decided not to discuss about Brazil, and hence did not use any of the suggested references.

 ref 16 is old - there have been more recent studies including e.g. Babar Z, Ramzan S et al. The Availability, Pricing, and Affordability of Essential Diabetes Medicines in 17 Low-, Middle-, and High-Income Countries. Frontiers in pharmacology. 2019;10(1375); Mukundiyukuri JP, Irakiza JJ, Nyirahabimana N et al. Availability, Costs and Stock-Outs of Essential NCD Drugs in Three Rural Rwandan Districts. Ann Glob Health. 2020;86:123 and Chow CK, Ramasundarahettige C, Hu W, AlHabib KF, Avezum A, Jr., Cheng X, et al. Availability and affordability of essential medicines for diabetes across high-income, middle-income, and low-income countries: a prospective epidemiological study. The lancet Diabetes & endocrinology. 2018;6(10):798-808 to name just a few Ref 16 has been removed and replaced in the discussion section. The replacement now reads as “An assessment of availability of essential NCD drugs in primary healthcare facilities in some Rwandan districts noted low availability of essential medicines for cardiovascular diseases management [13].”

 Similar comments to above - why mention diabetes when the paper is about hypertension - suggest broadening the title to talk about NCDs with a special emphasis on hypertension Thank you for your observation. Although the paper is about hypertension, diabetes is a common and related disease that may increase the risk of hypertension. We therefore assessed its technology as a possible screening tool in the primary healthcare facilities. 

 i would like to see comments on the ways forward to achieve the goals in the recent Nigerian Plan for NCDs and how these results can be used to provide guidance to the authorities, etc. Some key priority actions and activities from the Nigerian Plan for NCDs have been added under the discussion section.

---

## [Editor Report · Decision Letter 1]

19 Jan 2022

Essential medicines and technology for hypertension in primary healthcare facilities in Ebonyi State, Nigeria

PONE-D-21-19423R1

Dear Dr. Adeke,

We’re pleased to inform you that your manuscript has been judged scientifically suitable for publication and will be formally accepted for publication once it meets all outstanding technical requirements.

Kind regards,

Susan Horton

Academic Editor

PLOS ONE
---

## [Editor Report · Acceptance letter]

25 Jan 2022

PONE-D-21-19423R1 

Essential medicines and technology for hypertension in primary healthcare facilities in Ebonyi State, Nigeria 

Dear Dr. Adeke:

I'm pleased to inform you that your manuscript has been deemed suitable for publication in PLOS ONE. Congratulations! Your manuscript is now with our production department. 

Kind regards, 

on behalf of

Dr. Susan Horton 

Academic Editor

PLOS ONE